# Analysis of the Peripapillary and Macular Regions Using OCT Angiography in Patients with Schizophrenia and Bipolar Disorder

**DOI:** 10.3390/jcm10184131

**Published:** 2021-09-13

**Authors:** Edyta Koman-Wierdak, Joanna Róg, Agnieszka Brzozowska, Mario Damiano Toro, Vincenza Bonfiglio, Katarzyna Załuska-Ogryzek, Hanna Karakuła-Juchnowicz, Robert Rejdak, Katarzyna Nowomiejska

**Affiliations:** 1Department of General Ophthalmology, Medical University of Lublin, 20-079 Lublin, Poland; toro.mario@email.it (M.D.T.); robertrejdak@yahoo.com (R.R.); katarzyna.nowomiejska@umlub.pl (K.N.); 2Department of Psychiatry, Psychotherapy and Early Intervention, Medical University of Lublin, 20-079 Lublin, Poland; rog.joann@gmail.com (J.R.); karakula.hanna@gmail.com (H.K.-J.); 3Department of Mathematics and Medical Biostatistics, Medical University of Lublin, 20-079 Lublin, Poland; agnieszka.brzozowska@umlub.pl; 4Department of Ophthalmology, University of Zurich, 8091 Zurich, Switzerland; 5Department of Experimental Biomedicine and Clinical Neuroscience, Ophthalmology Section, University of Palermo, 90127 Palermo, Italy; enzabonfiglio@gmail.com; 6Department of Pathophysiology, Medical University of Lublin, 20-079 Lublin, Poland; katarzyna.zaluska-ogryzek@umlub.pl

**Keywords:** OCTA, schizophrenia, bipolar disorder, microvascular dysfunction

## Abstract

Purpose: To measure RNFL and vasculature around the optic disc and in the macula in patients with schizophrenia (SZ) and bipolar disorder (BD) using optical coherence tomography angiography (OCTA). Methods: 24 eyes of patients with SZ and 16 eyes of patients with BD as well as 30 eyes of healthy subjects were examined with OCTA. The radiant peripapillary capillary (RPC) density and RNFL thickness were measured in the peripapillary area. Moreover, macular thickness and vessel density were measured in both superficial and deep layers. Results: Significantly decreased values of vessel density in the macular deep vascular complex were found in the eyes of patients with SZ, compared to BD and the control group. The macular thickness in the whole vascular complex and in the fovea was significantly lower in SZ and BD group than in the control group. The radiant peripapillary vascular density and RNFL thickness were similar across groups. Conclusions: The retinal microvascular dysfunction occurs in the macula in patients with SZ and BD, but not around optic disc. OCTA can become an essential additional diagnostic tool in detection of psychiatric disorders.

## 1. Introduction

Schizophrenia (SZ) and bipolar disorder (BD) are mental diseases that typically present in late adolescence [1,2]. SZ is characterized by distortions in thinking, perception, emotions, language, sense of self and behaviour. In addition to psychiatric abnormalities, visual and ophthalmic impairments such as hypersensitivity to brightness, contrast and motions increases have also been reported in schizophrenia [1]. BD is defined by periods of abnormally elevated mood that last from days to weeks [2]. These multifactorial neurodevelopmental disorders have unclear pathogenesis due to the difficulty in direct assessment of their neuronal-level dysfunctions [3]. The neurobiology of SZ and BD is determined by reductions in grey matter and white matter as measured by structural neuroimaging [4,5]. However, using the current neuroimaging methods, it is difficult to draw unambiguous conclusions about the pathophysiology of mental diseases. Moreover, there are limitations in the brain imaging methods (low image resolution, cost, patient burden) [6]. Positron emission tomography demonstrated impaired circulation in schizophrenic patients, suggesting vascular involvement in this disease [7].

The retina develops from the same tissue (neuroectoderm) as the brain and shares many structural and functional similarities. Therefore, researchers have suggested that retinal changes can serve as a marker of progressive brain tissue loss [8,9,10]. This has prompted the use of a non-invasive retinal imaging technology, i.e., optical coherence tomography (OCT) and optical coherence tomography angiography (OCTA) [11,12,13,14,15]. They have been used in a wide spectrum in ophthalmology, to assess the thickness, volume, and structure of different layers of the retina [16,17,18,19,20]. In recent years, the structural analyses of the retina and retinal nerve fiber layer (RNFL) using OCT have been proven to be an excellent tool for detection and monitoring axonal loss and neurodegeneration in patients with neurodegenerative diseases including Alzheimer’s disease, Parkinson’s disease or multiple sclerosis [21,22,23,24]. In addition, previous studies have used OCT to assess changes in the macular and RNFL thickness in patients with SZ and BD [1,8,10,25,26,27]. Recent reports have provided evidence of retinal ganglion cell (GC) and RNFL atrophy, suggesting potential ocular biomarkers for SZ as a neurodegenerative disease process [1].

OCTA can measure the same parameters as OCT. Moreover, it facilitates en-face visualization of the retinal circulation into the superficial and deep vascular networks supplying the various retinal layers [28,29,30,31]. In addition, OCTA provides quantitative measurements of the retinal and optic disc vascular anatomy, permitting an objective approach for evaluating vascular pathology in disease [32,33,34]. However, it is a relatively new imaging method and literature on the structural retinal analysis by OCTA in patients with SZ and BD is scarce.

The aim of this study was to evaluate the vessel density, RNFL and macular thickness around the optic nerve head and in the macula using OCTA in patients with SZ and BD.

## 2. Material and Methods

The study was conducted between June 2019 and October 2020 at the Chair and Department of General and Pediatric Ophthalmology of the Medical University of Lublin, Poland. The study followed the tenets of the Declaration of Helsinki and a written approval of the Ethics Committee at the Medical University of Lublin, Poland was given (approval number KE-0254/247/2019).

The examination included 20 psychiatric patients (40 eyes), 12 patients (24 eyes) with SZ and 8 patients (16 eyes) with BD. The patients were recruited from the 1st Department of Psychiatry, Psychotherapy and Early Intervention of the Medical University of Lublin, Poland, according to the International Classification of Disease version 10 (ICD-10). Additionally, 15 patients (30 eyes), matched for age and gender, were recruited and served as a control group. Written informed consent was obtained from all patients. All participants of the study had received a comprehensive ophthalmological and psychiatric evaluation.

The ophthalmological examinations were performed at the Chair and Department of General and Pediatric Ophthalmology during the last week of hospitalization of the patients at the 1st Department of Psychiatry, Psychotherapy and Early Intervention of the Medical University of Lublin, Poland, when the criteria for symptomatic remission were fulfilled according to the ICD-10 classification for SZ and BD.

The exclusion criteria for all subjects were as follows: (1) systemic diseases (diabetes mellitus, cardiovascular disorders), (2) alcohol and other psychoactive substances dependence, (3) any eye disease (glaucoma, macular abnormalities, high myopia, ocular trauma, retinal detachment), (4) other diseases that may affect OCT or OCTA measurements.

Ophthalmological examinations were conducted among 15 healthy subjects recruited from medical students and hospital staff serving as a control group.

All patients underwent ophthalmological examinations performed by an ophthalmologist: the best-corrected visual acuity (BCVA) in the Snellen decimal scale, slit-lamp biomicroscopy, intra-ocular pressure (IOP) measurement and pupillary reactions.

OCTA examination was performed by an ophthalmologist using an Optovue (Fremont, CA, USA) device. The OCTA scans was calculated using the integrated software algorithm; 70,000 A-scan/s. Identification and segmentation methods of areas were selected automatically. The radiant peripapillary capillary (RPC) density and RNFL thickness were measured by OCTA in the peripapillary area (4.5 × 4.5 mm). Furthermore, macular thickness and vessel density (superficial and deep vascular complex) were measured with the same device on scans 6 × 6 mm for the macula. All scans were reviewed independently by two investigators (EKW and KN) to ensure correct segmentation and sufficient imaging quality, a cut-off quality score of OCTA scans was 7/10.

### Statistical Analysis

The database and statistical computations were carried out with STATISTICA 13.0 computer software (StatSoft, Kraków, Poland). The values of the analyzed measurable parameters were presented by the mean, median and standard deviation (SD) and as counts and percentage in the case of non-measurable parameters. Both eyes of each patient were tested, and the calculated means of the eye results were included in the statistical analysis (Appendix A). For the measurable features, the normal distribution of the analyzed parameters was assessed using the Shapiro–Wilk test. The Kruskal–Wallis test using multiple comparison analysis was used to compare multiple independent groups. A level of significance of *p* < 0.05 indicates the existence of statistically significant differences.

## 3. Results

### 3.1. Patient Group Characteristic

The study groups did not differ in age (Z = 1.59; *p* = 0.11). The mean age of the patients in the SZ group was 26.33 ± 5.26 (19–35 years), while in the BD group it was 24.13 ± 8.85 (15–43 years). The groups did not differ in terms of disease duration (Z = −0.59; *p* = 0.55). The duration of the disease in the SZ group was 42.58 ± 42.01 (Me = 36.00, ranging from 1 to 144 months) months, and in the BD group was 70.63 ± 116.01 (Me = 28.50, ranging from 4 to 366 months) months.

There was a significant relationship in the BD group between age and Whole capillary parameter (*p* = 0.03), the value of Whole capillary parameter decreases with age. Statistical analysis in the Whole capillary and gender showed significant differences in the group with SZ (*p* = 0.01), while no significant differences were found in the assessment of other parameters, nor in the BD group (*p* > 0.05).

In the SZ group, six patients were females and six were males. The mean body mass index (BMI) in the SZ group was 23 kg/m^2^. The mean BCVA was 0.95 (range 0.8–1.0; median 1.0) and the mean IOP was 14 mmHg (range 11–19 mmHg; median 14 mmHg). In the BD group there were five females and three males. The mean BMI was 24.7 kg/m^2^. The BCVA was 0.93 (range 0.7–1.0; median 1.0) and the mean IOP was 13 mmHg (min–max: 10–17 mmHg; median 14 mmHg).

In SZ groups, all the patients (*n* = 12) in the SZ group and 88% (*n* = 7) of the BD patients received second-generation antipsychotic (SGA) treatment. The mean olanzapine equivalents were: 23 (BD) and 36 (SZ). In the SZ group, 33% of the patients (*n* = 4) received selective serotonin reuptake inhibitors (SSRI) and/or typical antipsychotics treatment; 25% of the patients (*n* = 3) were taking benzodiazepines, 17% (*n* = 2) received mood stabilizers and 8% (*n* = 1) were administered a tricyclic antidepressant (TCA). All patients from the BD group were taking mood stabilizers (*n* = 8). 38% (*n* = 3) received SSRI and 25% (*n* = 2) were treated with typical antipsychotics.

Statistical analysis showed that SGA treatment was significantly more commonly used in SZ group than in the BD group (*p* = 0.01), mood stabilizers were more commonly used in BD group, than in SZ group (*p* = 0.0007).

The mean age in the control group was 26.8 years (Min–Max 23–35 years; median 24 years), the mean BCVA was 0.98 (Min–Max: 0.9–1.0; median 1.0) and the mean IOP was 14 mmHg (range 10–18 mmHg; median 13 mmHg). Eight of the normal subjects were females and seven subjects were males. The mean body mass index (BMI) in control group was 24 kg/m^2^. The pupillary reaction to the light, the anterior segment and the fundus of all included eyes were normal.

### 3.2. OCTA Analysis

The density of the whole capillary of the radial peripapillary capillary (4.5 × 4.5 mm) showed slight statistically significant differences (*p* = 0.02) between the groups. The values were only slightly lower in the SZ group in comparison to the eyes of the BD patients or the control. The assessment of RPC density parameters in individual quadrants (superior, nasal, inferior and temporal) between the groups did not differ significantly (*p* > 0.05) (Table 1 and Figure 1).

The values of the RNFL parameters (peripapillary, superior, nasal, inferior and temporal) in the peripapillary area (4.5 × 4.5 mm) did not differ significantly between the three groups (Table 2).

The statistical analysis of the macular vessel density (6 × 6 mm area) assessed as whole deep vascular complex showed significant differences between the three groups (*p* < 0.0001); (Figure 2). The values were significantly lower in the SZ group. The analysis of multiple comparisons showed differences between the SZ group and the control group (Z = 4.30; *p* = 0.00005) and between the group with SZ and BD (Z = 3.09; *p* = 0.006); there was no difference between the BD and control groups. However, no significant differences were found between the groups in the assessment of the other parameters of macular vessels density (Table 3).

Comparing the results of retinal thickness values measured in the whole macular area (6 × 6 mm) and fovea (automatically selected by the instrument software) we also observed highly significant differences between the groups: the whole macular area (*p* < 0.0001) and fovea (*p* = 0.02). The analysis of multiple comparisons showed that the whole parameter was significantly higher in the control group than in the SZ group (Z = 4.53; *p* = 0.00002) and compared to the group with BP disorder (Z = 5.03; *p* = 0.000001); however, no differences were found in the whole macular thickness assessment between SZ and BD. In the assessment of the fovea parameter, the analysis of multiple comparisons, revealed differences between the control group and BD (Z = 2.71; *p* = 0.02), the differences between the other groups were not statistically significant (Table 4).

## 4. Discussion

To the best of our knowledge, this study is the first to provide the retinal and peripapillary OCTA analysis in both groups of patients: with SZ and BD. We found significantly decreased macular parameters in patients with these two psychiatric diseases compared with the controls, especially whole (6 × 6 mm) macular thickness. Moreover, the reduction of the fovea thickness in the BP patients was statistically significant. We also observed reduced vascular density in the macular deep vascular plexus in the SZ patients.

OCTA was introduced in 2015 and since then, increasing amounts of research data are available. However, there is still an insufficient number of reports connecting OCTA examinations with neurodegenerative and psychiatric disorders. The pathophysiology of many disorders of the central nervous system (CNS) is related to altered microvasculature. OCTA has been used to describe vascular abnormalities in many neurological diseases, e.g., multiple sclerosis, Alzheimer’s disease, Leber hereditary optic neuropathy, Parkinson’s disease, Huntington’s disease, amyotrophic lateral sclerosis and migraine [35]. It appears that OCTA findings correlate quite well with the severity of the aforementioned diseases however they have received less attention in the area of mental illnesses such as SZ and BD.

Neuroimaging studies have illustrated impaired circulation in SZ patients, suggesting vascular involvement in this disease [7,36,37]. Assessment of retinal vascularization may play a significant role in analyzing the impact of recent episodes of disease and treatment on the long-term course of the disease. Our results of vascular impairment in the retina are in concordance with the results of the vascular impairment in the brain in patients with SZ showed with neuroimaging [7]. In Positron Emission Tomography of SZ patients, the blood flow was decreased in some areas of the brain and compensatorily increased in other regions of the brain. Moreover, it was proven that patients with SZ had highly visible naifold plexus, compared to unipolar, bipolar, and nonpsychiatric controls [36]. SZ patients with visible plexus had worse oculomotor performance, more negative symptoms, worse course, more severe illness, worse occupational functioning, and worse neuropsychological performance on tasks.

In the preliminary study conducted by Lizano and colleagues, [38] swept source OCTA was performed in 24 schizophrenia spectrum disorder (SSD) patients and 16 healthy controls. They assessed the vessel density, length, tortuosity, and diameter in the superficial vascular plexus by OCTA scans. The results showed no significant differences in the vessel density and length but significantly increased retinal vessel diameter and tortuosity, which was correlated with worse negative symptoms and poorer cognition in the SSD patients. The same results in the superficial vascular plexus were obtained in the SZ patients in our study, but additionally the density of the macular deep plexus was significantly reduced.

Macular thinning and macular volume have been assessed in many studies by OCT scans in SZ patients compared with control group. Several studies observed overall macular thinning in patients with SZ [39,40,41]. In an OCT study of the individual layers of the macula, Samani and colleagues found that total retinal thickness was significantly reduced (in the foveal and parafoveal regions), with significant thinning of the outer nuclear layer (ONL) and the inner segment layer (ISL) [39].

In a complete OCT analysis of the different retinal layers, a significant thinning of the inner sectors (RNFL, GCL, inner plexus layer (IPL) and inner nucleus layer (INL) was observed in the BD group compared with healthy subjects [42]. No significant changes in the outer retinal layers were found; moreover, the outer sectors (and volume) of INL were augmented in these two compared groups. These results indicate that the total macular thickness may not reflect the pathologic changes in the individual layers.

Macular thinning may be a marker of progressive neurodegeneration, which has been observed in both disorder [43,44]. Our results, demonstrating macular thinning, are similar to other studies in SZ patients [10,40,45]. The data about macular density in BD are rarer in the literature [42,44], although the evidence of neurodegenerative disorders suggests that macular thickness is correlated with disease progression and severity [46]. More studies on the physiopathology of retinal changes in patients with SZ and BD should be conducted to elucidate the differences in vascular density and macular thickness.

Surprisingly and contrary to what was expected, this study did not detect any significant differences in the peripapillary RNFL thickness between the SZ and BD patients and the healthy controls. Moreover, the mean RNFL thickness parameters in the peripapillary area (4.5 × 4.5 mm) and superior and temporal quadrants were higher in the SZ/BD patients than in the control group. These results suggest that undetectable loss of unmyelinated axons creating the optic nerve is possible in patients with exacerbation of symptoms requiring hospitalization. To explain these findings, Ascaso et al. [40] hypothesized that neuro-inflammation, which can occur during acute episodes, may increase RNFL thickness, thus masking the effects of axonal loss and thinning in RNFL that are seen in chronic schizophrenia and BD patients.

Other studies showed that there were no differences in RNFL thickness in the “recent illness” group vs. controls or between treatment responsive patients and controls [40,46]. Additionally, RNFL thinning was associated with worse clinical symptom severity in one study [26], and no association was reported in several studies [42,46,47,48]. Lizano and colleagues [27] observed significant thinning of the peripapillary RNFL in OCT scans of the overall and nasal region in SZ and BD groups, but their meta-analysis involved a mix of acutely and chronically impaired patients (outpatients and inpatients). The findings in the study conducted by Ascaso et al., 2015 [40], confirm the results described in their first report showing a reduced peripapillary RNFL thickness, macular inner ring thickness and macular volume in SZ patients vs. controls with the use of OCT. However, the differences in retinal parameters were only significant in patients with “no-recent illness episode” but not in “recent illness episode” patients. Differences in the same parameters were reported by Lee et al. [9,10] in “chronic” and “long-term chronic” patients with SZ, but not in “acute” patients. Alizedah and co-workers [49] compared RNFL thickness results (measured with OCT) between individuals with acute or chronic SSD and healthy controls. Disease duration and RNFL thickness were linearly associated, depending on the chronic or acute state. In participants with acute SSD was associated with thicker RNFL, while in participants with a chronic SSD was associated with a thinner RNFL. Asanad and colleagues [50] conducted the study that evaluated the potential diagnostic OCT indexes (in both inner and outer retina) in SSD patients. The peripapillary RNFL was significantly thinner, primarily of the temporal and inferior quadrants, in SSD relative to controls. There were no significant abnormalities in macular RNFL, retinal ganglion cell and photoreceptor complexes however peripapillary RNFL, central macula and outer photoreceptor complex thicknesses were together able to predict SSD patients with high sensitivity (80%) and specificity (71%).

On the other hand, there is sample evidence of a decrease in the RNFL thickness in the macular and peripapillary area in patients with SZ and BD [8,10,27,40] which suggests that these findings may be a consequence of the progression of neurodegenerative changes that are measurable in the retina.

In these circumstances, the reduction of peripapillary RNFL thickness and macular volume associated with SZ and BD may not be observed due to the acute neuroinflammatory process [51]. In patients without a recent episode of illness, changes suggesting axonal degeneration may be apparent [10,40]. Future longitudinal studies should assess the presence of neuroinflammatory markers to test this hypothesis and their relationship with OCTA results. There are several limitations associated with this study. The small sample sizes, difficulties in interpretation and no unanimous protocols (on parameters that should be taken into account) limit the level of validity of the findings and may lead to false-negative or -positive results. However, in other studies, the sample sizes were similar, for example, 28 patients with SZ in the recent study by Silverstein and colleagues [52], 30 patients with SSD in the study of Alizedah and colleagues [49] and 22 patients in the study of Budakoglu and colleagues [53]. The small sample size corresponded to the lack of differentiation between the gender, disease duration and type of medications taken. The majority of our patients were receiving antipsychotic medication at the time of data collection, and it is impossible to exclude the potential neuroprotective effects of these drugs [54], which may have obscured the minor differences in RNFL between the groups. Furthermore, anatomical variables such as individual differences in blood flow and anatomy of vessels may cause variability between subjects and results [55].

Our results did not reflect the differences in disease duration or medication status. A few studies reported a significantly negative relationship between RNFL thickness and disease duration [8,10,26,47,56] while others did not observe a relationship [40,42,51,57].

## 5. Conclusions

OCTA, providing valuable data about the structural changes of RNFL, macular thickness and retinal vascular network, can become an essential tool in detection and monitoring of a wide range of neurodegenerative disorders. It can be used for the development of biomarkers to monitor the disease progression and develop potential future treatments in SZ and BD. Larger prospective longitudinal studies are needed to detect whether and in which stage of the disease progressive retinal changes occur to optimize the quality of OCTA images and accurate interpretation of the data.

## Figures and Tables

**Figure 1 jcm-10-04131-f001:**
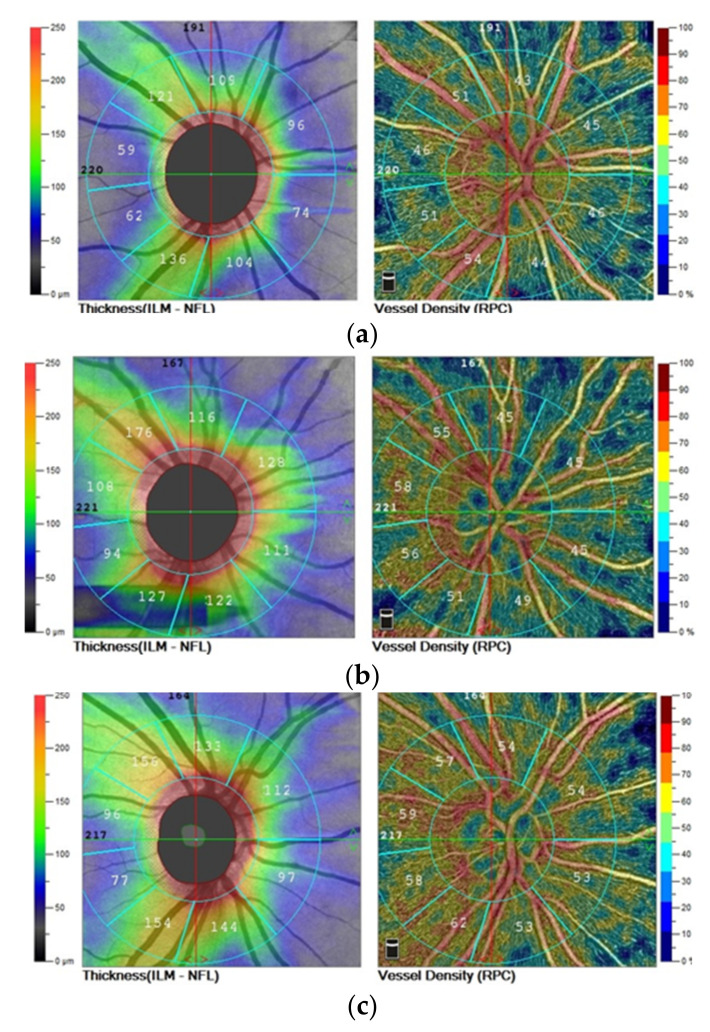
OCTA of nerve fiber layer thickness and the radial peripapillary capillary density in the right eye of patient with (**a**) schizophrenia, (**b**) bipolar disorder, (**c**) control group.

**Figure 2 jcm-10-04131-f002:**
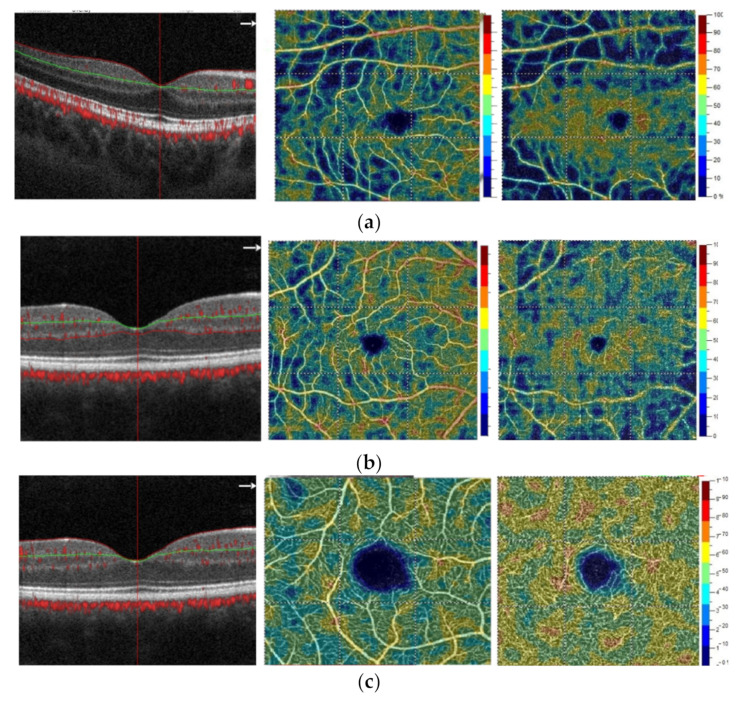
OCTA of the macular vessel density—B-scan, superficial vascular complex, deep vascular complex—in the right eye of patient with (**a**) schizophrenia, (**b**) bipolar disorder, (**c**) control group.

**Table 1 jcm-10-04131-t001:** Radial peripapillary capillary (RPC) density (%) in the peripapillary area in the eyes of the SZ and BD patients and in the control group.

RPC Density (%)	SZ			BD			Control Group			Statistical Analysis
	Mean	Median	SD	Mean	Median	SD	Mean	Median	SD	
Whole capillary	48.09	49.20	3.09	49.99	50.65	1.95	50.10	50.50	2.63	H = 7.86 *p* = 0.02 *
Superior	51.71	52.00	6.02	51.44	51.00	5.99	52.90	53.00	3.38	H = 0.75 *p* = 0.69
Nasal	52.58	52.00	7.76	52.56	51.50	5.73	53.77	53.50	5.30	H = 0.78 *p* = 0.68
Inferior	52.42	51.50	5.14	52.06	52.00	4.55	54.10	54.00	3.71	H = 3.77 *p* = 0.15
Temporal	50.58	51.50	6.82	50.50	50.50	5.79	52.17	53.00	4.34	H = 0.97 *p* = 0.61

Abbreviations: SZ—schizophrenia; bipolar disorder—BD; H—value of Kruskall-Wallis test; SD standard deviation; *—statistical significance.

**Table 2 jcm-10-04131-t002:** RNFL parameters in the optic nerve disc area in the eyes of the SZ and BP patients and in the control group.

RNFL µm	SZ			BD			Control Group			Statistical Analysis
	Mean	Median	SD	Mean	Median	SD	Mean	Median	SD	
Peripapillary	118.09	114.00	23.56	113.07	114.00	20.45	112.57	116.00	9.01	H = 0.98 *p* = 0.61
Superior	137.17	138.00	20.10	140.00	138.00	28.05	132.63	135.00	10.06	H = 2.08 *p* = 0.35
Nasal	97.42	98.50	18.72	96.69	96.00	15.72	100.14	99.00	11.47	H = 0.70 *p* = 0.71
Inferior	142.63	140.50	20.78	138.31	132.00	29.41	140.20	146.00	13.15	H = 0.57 *p* = 0.75
Temporal	79.04	76.50	12.84	76.69	68.50	16.78	76.60	79.00	11.16	H = 0.90 *p* = 0.64

Abbreviations: SZ—schizophrenia; bipolar disorder—BD; H—value of Kruskall-Vallis test; SD—standard deviation.

**Table 3 jcm-10-04131-t003:** Vessel density in the macular superficial and deep vascular complexes in two areas (whole 6 × 6 mm area and in the fovea) in the SZ and BD patients and the control group.

Macula Vessel Density %	SZ			BD			Control Group			Statistical Analysis
	Mean	Median	SD	Mean	Median	SD	Mean	Median	SD	
Whole superficial vascular complex	47.93	48.40	3.37	48.56	48.60	3.13	47.80	48.10	2.84	H = 0.55 *p* = 0.76
Whole deep vascular complex	43.66	42.45	5.50	50.41	49.25	6.54	50.66	51.45	4.26	H = 19.91 *p* < 0.0001 *
Foveal superficial vascular complex	24.15	24.95	6.96	23.59	21.45	9.45	21.27	20.75	3.86	H = 3.00 *p* = 0.22
Foveal deep vascular complex	38.81	40.40	7.74	39.38	37.40	8.64	38.43	39.00	4.87	H = 1.15 *p* = 0.56

Abbreviations: SZ—schizophrenia; bipolar disorder—BD; H—value of Kruskall-Wallis test; SD—standard deviation; *—statistical significance.

**Table 4 jcm-10-04131-t004:** Macular thickness in the whole area (6 × 6 mm) and in the fovea in the three groups of patients.

Macular Thickness µm	SZ			BD			Control Group			Statistical Analysis
	Mean	Median	SD	Mean	Median	SD	Mean	Median	SD	
Whole	284.25	283.00	14.77	278.50	273.50	12.59	305.23	301.50	11.33	H = 33.05 *p* < 0.0001 *
Fovea	260.67	262.00	24.28	249.63	244.00	21.43	263.53	263.50	15.37	H = 7.41 *p* = 0.02 *

Abbreviations: SZ—schizophrenia; bipolar disorder—BD; H—value of Kruskall-Wallis test; SD—standard deviation; *—statistical significance.

## Data Availability

Data are available on reasonable request to the corresponding author.

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
