# Peer review of "Analysis of the Peripapillary and Macular Regions Using OCT Angiography in Patients with Schizophrenia and Bipolar Disorder"

_jcm, 2021, doi:10.3390/jcm10184131_

Round 1

Reviewer 1 Report

The authors investigated the peripapillary and macular vascular change in patients with SZ and BD using OCTA. This is an interesting study, which nevertheless raises some concerns:

  1. Although the authors described in the Discussion, the biggest limitation of this study is that the number of patient groups is too small. I do not know how a group of 20 patients (12 SZ, 8 BD) could represent the characteristics of the disease. There are many variables such as age, gender, illness period, medications used, patient status, etc. There is a great deal of doubt as to whether the measurement results of 20 patients can be representative.
  2. Table 1 shows the statistical significance of RPC density with p=0.02 for the whole capillary. However, none of the remaining 4 directions (sup, nasal, inf. Temp) showed statistical significance. I wonder if there really is a difference. It is considered that there is a possibility of statistical error due to the small number of patient groups.

  3. The capillary vessel density measurement method is described only in writing. This can make it difficult to understand exactly how it was measured. I think it would be good to attach a figure showing the actual OCT and OCTA pictures of each measurement area as examples, indicating that this area was measured.

  4. What does “SSD” on Page 8, line 190 mean?

Author Response

Reviewer 1     

The authors investigated the peripapillary and macular vascular change in patients with SZ and BD using OCTA. This is an interesting study, which nevertheless raises some concerns:

  1. Although the authors described in the Discussion, the biggest limitation of this study is that the number of patient groups is too small. I do not know how a group of 20 patients (12 SZ, 8 BD) could represent the characteristics of the disease. There are many variables such as age, gender, illness period, medications used, patient status, etc. There is a great deal of doubt as to whether the measurement results of 20 patients can be representative.

In the results section the analysis in regard to age, illness period and medications used.

In the results section (line 94) it has been added:

The study groups did not differ in age (Z = 1.59; p = 0.11). The mean age of patients in the SZ group was 26.33 ± 5.26 (19-35 years), while in the BD group it was 24.13 ± 8.85 (15-43 years). The groups did not differ in terms of disease duration (Z = -0.59; p = 0.55). The duration of the disease in the SZ was 42.58 ± 42.01 (Me = 36.00, range from 1 to 144 months) months, and in the BD group was 70.63 ± 116.01 (Me = 28.50, range from 4 to 366 months) months.

There was a significant relationship in the BD group between age and Whole capillary parameter( p=0.03), the value of Whole capillary parameter decreases with age. Statistical analysis in the Whole capillary and gender showed significant differences in the group with SZ  (p = 0.01), while no significant differences were found in the assessment of other parameters, also in the BD group (p> 0.05).

Statistical analysis showed that SGA treatment was significantly more commonly used in SZ group than in the BD group (p=0.01), mood stabilizers were more commonly used in BD group, than in SZ group (p=0.0007).

In the discussion chapter it has been added:

Line 190 : In the preliminary study of Lizano et collegues [21] Swept Source OCTA was performed in 24 schizophrenia spectrum disorder (SSD) patients and 16 healthy controls.

Line 257: However, in other studies the sample sizes were similar, for example 28 patients with SZ in the recent study by Silverstein and colleagues [36], 30 patients with SSD in the study of Alizedah and coworkers [33] and 22patients in the study of Budakoglu and colleagues [37]

The following references have been added:

Silverstein SM, Lai A, Green KM, Crosta C, Fradkin SI, Ramchandran RS. Retinal Microvasculature in Schizophrenia. Eye Brain. 2021 Jul 24;13:205-217.

Alizadeh M, Delborde Y, Ahmadpanah M, Seifrabiee MA, Jahangard L, Bazzazi N, Brand S. Non-linear associations between retinal nerve fibre layer (RNFL) and positive and negative symptoms among men with acute and chronic schizophrenia spectrum disorder. J Psychiatr Res. 2021 Sep;141:81-91.

Budakoglu O, Ozdemir K, Safak Y, Sen E, Taskale B. Retinal nerve fibre layer and peripapillary vascular density by optical coherence tomography angiography in schizophrenia. Clin Exp Optom. 2021 Sep;104(7):788-794.

2.Table 1 shows the statistical significance of RPC density with p=0.02 for the whole capillary. However, none of the remaining 4 directions (sup, nasal, inf. Temp) showed statistical significance. I wonder if there really is a difference. It is considered that there is a possibility of statistical error due to the small number of patient groups.

As this result is only slightly significant, the following sentence has been removed from the abstract (line 8):

Values of the whole capillary peripapillary density were significantly lower in the group with SZ in comparison to the eyes with BD or control.

In the results of the abstract it has been added:

The radiant peripapillary vascular density and RNFL thickness were similar across groups.

The conclusion of the abstract has been changed into (line 11):

The retinal microvascular dysfunction occurs in the deep vascular complex of the macula in patients with SZ and BD, but not around optic disc.

In the results section it is now written (line 126):

The density of whole capillary of the radial peripapillary capillary (4.5x4.5mm) showed slight statistically significant differences (p = 0.02) between groups. Values were only slightly lower in the group with SZ in comparison to the eyes with BD or control.

  1. The capillary vessel density measurement method is described only in writing. This can make it difficult to understand exactly how it was measured. I think it would be good to attach a figure showing the actual OCT and OCTA pictures of each measurement area as examples, indicating that this area was measured.

The following OCTA pictures were attached in part 3.2 OCTA analysis:

Figure 1. OCTA of nerve fiber layer thickness and the radial peripapillary capillary density in the right eye of patient with:

  1. schizophrenia

  1. bipolar disorder

  1. control group

Figure 2. OCTA of macular vessel density- B-scan, superficial vascular complex, deep vascular complex- in the right eye of patient with:

  1. schizophrenia

  1. bipolar disorder

  1. control group

What does “SSD” on Page 8, line 190 mean?

It was added in the text (line 190): In the preliminary study of Lizano et collegues [21] Swept Source OCTA was performed in 24 schizophrenia spectrum disorder (SSD) patients and 16 healthy controls.

Reviewer 2 Report

This is an overall well designed and interesting study comparing structural and vascular parameters in eyes of patients with schizophrenia (SZ) and bipolar disorder (BD) to those of healthy subjects, evaluated using OCTA.

While several studies in the past found decreased retinal thickness in neurodegenerative disorders, studies evaluating retinal vasculature with OCTA in SZ and BD are scarcer (Budakoglu O, Ozdemir K, Safak Y, Sen E, Taskale B. Retinal nerve fibre layer and peripapillary vascular density by optical coherence tomography angiography in schizophrenia. Clin Exp Optom. 2021 Feb 25:1-7; Silverstein SM, Lai A, Green KM, Crosta C, Fradkin SI, Ramchandran RS. Retinal Microvasculature in Schizophrenia. Eye Brain. 2021 Jul 24;13:205-217.). Neuroimaging studies have demonstrated impaired circulation in schizophrenic patients, suggesting vascular involvement in this disease (Andreasen NC, Calarge CA, O’Leary DS. Theory of mind and schizophrenia: a positron emission tomography study of medication-free patients. Schizophr Bull 2008; 34: 708–719.;

Curtis CE, Iacono WG, Beiser M. Relationship between nailfold plexus visibility and clinical, neuropsychological, and brain structural measures in schizophrenia. Biol Psychiatry 1999; 46: 102–109.; Uranova NA, Zimina IS, Vikhreva OV, et al. Ultrastructural damage of capillaries in the neocortex in schizophrenia. World J Biol Psychiatry 2010; 11: 567–578.). Authors should consider commenting more extensively their results focusing on vascular parameters (although they are a bit controversial and not so statistically relevant as structural data), in relation to previously published data.

Please find below additional specific comments and suggestions:

  • Line 58: please consider adding references Asanad S, O'Neill H, Addis H, et al. Neuroretinal Biomarkers for Schizophrenia Spectrum Disorders. Transl Vis Sci Technol. 2021;10(4):29. doi:10.1167/tvst.10.4.29 and Alizadeh M, Delborde Y., Ahmadpanah M., eta al. Non-linear associations between retinal nerve fibre layer (RNFL) and positive and negative symptoms among men with acute and chronic schizophrenia spectrum disorder, Journal of Psychiatric Research. 2021; 141:81-91. Results of these recent papers should be also discussed in relation to presented findings in the discussion section.
  • Was the vascular density of OCTA scans calculated using the integrated software algorithm? Please provide details including details of specific acquisitions (e.g. # of A scans for each scan) , identification methods of areas as shown in table 1,2 (superior, nasal, temporal, inferior) and segmentation methods.
  • Is there a cut-off quality score that has been chosen to include OCTA scans in the dataset?
  • Please, check BCVA data as mean BCVA is probably not 1.0 snellen equivalent if anyone of the included cases had less than 1.0 VA
  • Consider reporting p values for comparisons of demographic data between study groups
  • Lines 161-168; please indicate more clearly that this section show comparison results of retinal thickness values measured in the whole macular area and fovea. Please also specify if these areas (macula and fovea) were automatically selected by the instrument software or manually
  • Table 3: please indicate the precise meaning of terms “whole superficial” and “whole deep” (superficial & deep vascular complexes?)
  • The term “superficial capillary layer” should be replaced with “superficial vascular plexus”

The formal quality of presentation could be implemented throughout the whole text, including English expression. Just to mention some parts that would benefit of formal revision:

  • Lines 47-50: please rephrase including a main clause and subordinates
  • Lines 65-66: please check English
  • Line 190: please spell out “SSD”
  • Discussion clarity should be implemented (English expression and logic flow)

Author Response

Rewiewer 2

This is an overall well designed and interesting study comparing structural and vascular parameters in eyes of patients with schizophrenia (SZ) and bipolar disorder (BD) to those of healthy subjects, evaluated using OCTA.

While several studies in the past found decreased retinal thickness in neurodegenerative disorders, studies evaluating retinal vasculature with OCTA in SZ and BD are scarcer (Budakoglu O, Ozdemir K, Safak Y, Sen E, Taskale B. Retinal nerve fibre layer and peripapillary vascular density by optical coherence tomography angiography in schizophrenia. Clin Exp Optom. 2021 Feb 25:1-7; Silverstein SM, Lai A, Green KM, Crosta C, Fradkin SI, Ramchandran RS. Retinal Microvasculature in Schizophrenia. Eye Brain. 2021 Jul 24;13:205-217.).

In the discussion chapter it has been added (line 257):

However, in other studies the sample sizes were similar, for example 28 patients with SZ in the recent study by Silverstein and colleagues [36], 30 patients with SSD in the study of Alizedah and coworkers [33] and 22 patients in the study of Budakoglu and colleagues [37].

Neuroimaging studies have demonstrated impaired circulation in schizophrenic patients, suggesting vascular involvement in this disease (Andreasen NC, Calarge CA, O’Leary DS. Theory of mind and schizophrenia: a positron emission tomography study of medication-free patients. Schizophr Bull 2008; 34: 708–719.; Curtis CE, Iacono WG, Beiser M. Relationship between nailfold plexus visibility and clinical, neuropsychological, and brain structural measures in schizophrenia. Biol Psychiatry 1999; 46: 102–109.; Uranova NA, Zimina IS, Vikhreva OV, et al. Ultrastructural damage of capillaries in the neocortex in schizophrenia. World J Biol Psychiatry 2010; 11: 567–578.).

In the introduction chapter it has been added (line 26):

Positron emission tomography demonstrated impaired circulation in schizophrenic patients, suggesting vascular involvement in this disease [7].

Authors should consider commenting more extensively their results focusing on vascular parameters (although they are a bit controversial and not so statistically relevant as structural data), in relation to previously published data.

In the discussion chapter it has been added (line 182):

Our results of vascular impairment in the retina are in concordance with the results of the vascular impairment in the brain in patients with SZ showed with neuroimaging [7]. In Positron Emission Tomography of SZ patients the blood flow was decreased in some areas of the brain and compensatory increased in other regions of the brain. Moreover, it was proven that patients with SZ had highly visible naifold plexus compared to unipolar, bipolar, and nonpsychiatric controls [19]. SZ patients with visible plexus had worse oculomotor performance, more negative symptoms, worse course, more severe illness, worse occupational functioning, and worse neuropsychological performance on tasks.

The following reference have been added:

Andreasen NC, Calarge CA, O'Leary DS. Theory of mind and schizophrenia: a positron emission tomography study of medication-free patients. Schizophr Bull. 2008 Jul;34(4):708-19.

Curtis CE, Iacono WG, Beiser M. Relationship between nailfold plexus visibility and clinical, neuropsychological, and brain structural measures in schizophrenia. Biol Psychiatry. 1999 Jul 1;46(1):102-9.

Please find below additional specific comments and suggestions:

  • Line 58: please consider adding references Asanad S, O'Neill H, Addis H, et al. Neuroretinal Biomarkers for Schizophrenia Spectrum Disorders. Transl Vis Sci Technol. 2021;10(4):29. doi:10.1167/tvst.10.4.29 and Alizadeh M, Delborde Y., Ahmadpanah M., eta al. Non-linear associations between retinal nerve fibre layer (RNFL) and positive and negative symptoms among men with acute and chronic schizophrenia spectrum disorder, Journal of Psychiatric Research. 2021; 141:81-91. Results of these recent papers should be also discussed in relation to presented findings in the discussion section.

In the discussion section it has been added (line 241): Asanad and collegues conducted the study that evaluated the potential diagnostic OCT  indexes (in both inner and outer retina) in SSD patients. The peripapillary RNFL was significantly thinner, primarily of the temporal and inferior quadrants, in SSD relative to controls. There were no significant abnormalities in macular RNFL, retinal ganglion cell and photoreceptor complexes however peripapillary RNFL, central macula and outer photoreceptor complex thicknesses were together able to predict SSD patients with high sensitivity (80%) and specificity(71%).

The following reference has been added:

Asanad S, O'Neill H, Addis H, Chen S, Wang J, Goldwaser E, Kochunov P, Hong LE, Saeedi OJ. Neuroretinal Biomarkers for Schizophrenia Spectrum Disorders. Transl Vis Sci Technol. 2021 Apr 1;10(4):29.

In the discussion section it has been added (line 237): Alizedah and coworkers compared RNFL thickness results (measured with OCT) between individuals with acute or chronic SSD and healthy controls. Disease duration and RNFL thickness were linearly associated, depending on the chronic or acute state. In participants with acute SSD was associated with thicker RNFL, while in participants with a chronic SSD was associated with a thinner RNFL.

The following reference has been added:

Alizadeh M, Delborde Y, Ahmadpanah M, Seifrabiee MA, Jahangard L, Bazzazi N, Brand S. Non-linear associations between retinal nerve fibre layer (RNFL) and positive and negative symptoms among men with acute and chronic schizophrenia spectrum disorder. J Psychiatr Res. 2021 Sep;141:81-91.

  • Was the vascular density of OCTA scans calculated using the integrated software algorithm? Please provide details including details of specific acquisitions (e.g. # of A scans for each scan) , identification methods of areas as shown in table 1,2 (superior, nasal, temporal, inferior) and segmentation methods.

In the material and methods section it has been added (line 76) : The OCTA scans was calculated using the integrated software algorithm; 70 000 A-scan /s. Identification and segmentation methods of areas was selected automatically.

  • Is there a cut-off quality score that has been chosen to include OCTA scans in the dataset?

In the material and methods section it has been added (line 82) : A cut-off quality score in OCTA scans was 7/10.

  • Please, check BCVA data as mean BCVA is probably not 1.0 snellen equivalent if anyone of the included cases had less than 1.0 VA

It was checked and changed respectively for SZ, BD and control group.

In the Results section it has been added (line 105) : BCVA- 0.95, 0.93 and 0.98.

  • Consider reporting p values for comparisons of demographic data between study groups

In the results section (line 94) it has been added:

The study groups did not differ in age (Z = 1.59; p = 0.11). The mean age of patients in the SZ group was 26.33 ± 5.26 (19-35 years), while in the BD group it was 24.13 ± 8.85 (15-43 years). The groups did not differ in terms of disease duration (Z = -0.59; p = 0.55). The duration of the disease in the SZ was 42.58 ± 42.01 (Me = 36.00, range from 1 to 144 months) months, and in the BD group was 70.63 ± 116.01 (Me = 28.50, range from 4 to 366 months) months.

There was a significant relationship in the BD group between age and Whole capillary parameter( p=0.03), the value of Whole capillary parameter decreases with age. Statistical analysis in the Whole capillary and gender showed significant differences in the group with SZ  (p = 0.01), while no significant differences were found in the assessment of other parameters, also in the BD group (p> 0.05).

  • Lines 161-168; please indicate more clearly that this section show comparison results of retinal thickness values measured in the whole macular area and fovea. Please also specify if these areas (macula and fovea) were automatically selected by the instrument software or manually

In the results section it has been added (line 151):

Comparing results of retinal thickness values measured in the whole macular area (6x6mm) and fovea (automatically selected by the instrument software) we also observed a highly significant differences between the groups: the whole macular area (p <0.0001) and fovea (p = 0.02).

  • Table 3: please indicate the precise meaning of terms “whole superficial” and “whole deep” (superficial & deep vascular complexes?)

It was amended correctly in the Table 3 as follows (line 148): whole/ foveal superficial vascular complex and whole/ foveal deep vascular complex

  • The term “superficial capillary layer” should be replaced with “superficial vascular plexus”

It was replaced in whole text with superficial/ deep vascular plexus .

The formal quality of presentation could be implemented throughout the whole text, including English expression. Just to mention some parts that would benefit of formal revision:

  • Lines 47-50: please rephrase including a main clause and subordinates

These sentence have been corrected as follows (line 29): The retina develops from the same tissue (neuroectoderm) as the brain and shares many structural and functional similarities. Therefore, researchers have suggested that retinal changes can serve as a marker of progressive brain tissue loss.

  • Lines 65-66: please check English

This sentence has been corrected as follows (line 45): However, it is relatively a new imaging method and literature on the structural retinal analysis by OCTA in patients with SZ and BD is scarce.

  • Line 190: please spell out “SSD”

It was added in line 190: In the preliminary study of Lizano et collegues [21] Swept Source OCTA was performed in 24 schizophrenia spectrum disorder (SSD) patients and 16 healthy controls.  

  • Discussion clarity should be implemented (English expression and logic flow)

English language in the manuscript has been corrected by a professional.

Round 2

Reviewer 1 Report

I agree with the authors' corrections. However, I think the small number of patients is an unavoidable handicap.